Influence of age and sex on winter site fidelity of sanderlings Calidris alba

Lourenço Pedro M. p.m.g.lourenco@gmail.com 1
Alves José A. 2 3
Reneerkens Jeroen 4
Loonstra AH Jelle 4
Potts Peter M. 5
Granadeiro José P. 1
Catry Teresa 1
1 Centro de Estudos do Ambiente e do Mar (CESAM), Departamento de Biologia Animal, Faculdade de Ciências, Universidade de Lisboa , Lisboa , Portugal
2 Centro de Estudos do Ambiente e do Mar (CESAM), Universidade de Aveiro , Aveiro , Portugal
3 South Iceland Research Centre, University of Iceland , Selfoss , Iceland
4 Conservation Ecology Group, Groningen Institute for Evolutionary Life Sciences (GELIFES), University of Groningen , Groningen , The Netherlands
5 Farlington Ringing Group , Southampton , Hampshire , United Kingdom
Gandini Patricia
Electronic publication date: 2016 Sep 28
Publication date: 2016
Volume: 4
Electronic Location ID: e2517
Received 2016 Jun 8; Accepted 2016 Sep 1
Copyright: ©2016 Lourenço et al.
Copyright year: 2016
Copyright holder: Lourenço et al.
License: This is an open access article distributed under the terms of the Creative Commons Attribution License, which permits unrestricted use, distribution, reproduction and adaptation in any medium and for any purpose provided that it is properly attributed. For attribution, the original author(s), title, publication source (PeerJ) and either DOI or URL of the article must be cited.
License URL: https://creativecommons.org/licenses/by/4.0/

Keywords: Site fidelity, Shorebirds, Estuaries, Sex, Age, Movement, Distance, Winter, Foraging areas

Funding: Fundação para a Ciência e Tecnologia FCT/MEC through national funds FEDER Waddenfonds WF-209925 Pedro M. Lourenço (SFRH/BPD/84237/2012), Teresa Catry (SFRH/BPD/102255/2014) and José A. Alves (SFRH/BPD/91527/2012) benefited from post-doctoral grants from Fundação para a Ciência e Tecnologia. We received financial support to CESAM (UID/AMB/50017), from FCT/MEC through national funds, and the co-funding by the FEDER, within the PT2020 Partnership Agreement and Compete 2020. Jeroen Reneerkens was financially supported by Waddenfonds (project Metawad, WF-209925). The funders had no role in study design, data collection and analysis, decision to publish, or preparation of the manuscript.

==============================
Many migratory bird species show high levels of site fidelity to their wintering sites, which confers advantages due to prior knowledge, but may also limit the ability of the individual to move away from degrading sites or to detect alternative foraging opportunities. Winter site fidelity often varies among age groups, but sexual differences have seldom been recorded in birds. We studied a population of individually colour-marked sanderlings wintering in and around the Tejo estuary, a large estuarine wetland on the western coast of Portugal. For 160 individuals, sighted a total of 1,249 times between November 2009 and March 2013, we calculated the probability that they moved among five distinct wintering sites and how this probability is affected by distance between them. To compare site fidelity among age classes and sexes, as well as within the same winter and over multiple winters, we used a Site Fidelity Index (SFI). Birds were sexed using a discriminant function based on biometrics of a large set of molecularly sexed sanderlings (n = 990). The vast majority of birds were observed at one site only, and the probability of the few detected movements between sites was negatively correlated with the distance among each pair of sites. Hardly any movements were recorded over more than 15 km, suggesting small home ranges. SFI values indicated that juveniles were less site-faithful than adults which may reflect the accumulated knowledge and/or dominance of older animals. Among adults, females were significantly less site faithful than males. A sexual difference in winter site fidelity is unusual in shorebirds. SFI values show site-faithfulness is lower when multiple winters were considered, and most birds seem to chose a wintering site early in the season and use that site throughout the winter. Sanderlings show a very limited tendency to explore alternative wintering options, which might have implications for their survival when facing habitat change or loss (e.g., like severe beach erosion as can be the case at one of the study sites).

Introduction

Although migratory birds are extremely mobile, they are often remarkably site-faithful to their breeding (e.g., Harvey, Greenwood & Perrins, 1979; Jackson, 1994), wintering (e.g., Burton & Evans, 1997; Catry et al., 2003; Leyrer et al., 2006) and staging sites (Gudmundsson & Lindström, 1992; Kruckenberg & Borbach-Jaene, 2004; Loonstra, Piersma & Reneerkens, 2016), both within the same season and among years. Such high levels of site fidelity are likely to confer advantages related to prior ownership of territories, previous knowledge of foraging locations, potential nest sites and local predators, and maintenance of social position within local dominance structures (Greenwood & Harvey, 1982; Alerstam, 1990; Snell-Rood & Cristol, 2005).

However, when confronted with a rapidly changing environment, the regular use of the same set of sites over the years may expose individuals to increasingly poorer conditions and habitats (Battin, 2004), when a traditionally used site suffers negative changes over time (e.g., Porzig et al., 2014) e.g., through a limitation in the use of all available habitat (Matthiopoulos, Harwood & Thomas, 2005). Therefore, the ability of animals to disperse and/or sample new areas can be of critical importance for highly site faithful species, under the current fast pace of global environmental change. Unfortunately, evaluating the use of multiple sites over large areas by the same individuals requires intensive effort to repeatedly detect marked individuals, which hinders the possibility of expanding efforts over large areas and impedes the detection of dispersion over longer distances. In addition, the traditionally low rate of movement among sites hinders the use of remote tracking techniques as the very large sample sizes needed to detect rare movements would greatly increase the cost of such an endeavour (e.g., Nathan et al., 2003; Hobson, 2005).

Site fidelity may be influenced by age, as juvenile animals of most species exhibit dispersive and/or exploratory behaviours during which they search for places to ultimately settle (Clobert et al., 2001). However, few studies have focused on age effects in winter site fidelity in migratory birds, and the available studies have reported conflicting results in this issue. Juvenile western sandpipers Calidris mauri showed larger home ranges and weaker homing behaviour than adults in winter (Warnock & Takekawa, 1996; Baccetti et al., 1999), and younger white-fronted geese Anser albifrons were more likely to move among wintering sites than older birds (Wilson et al., 1991). However, there were no differences in winter site fidelity among different age groups of either pink-footed geese Anser brachyrhynchus (Fox et al., 1994) or American black ducks Anas rubripes (Diefenbach, Nichols & Hines, 1988).

Sex can also influence site fidelity. Most research on site fidelity in migratory birds has focused on the breeding season, generally showing that males tend to be more site-faithful and disperse over shorter distances between years than females (Clarke, Sæther & Røskaft, 1997; Gunnarsson et al., 2012). Such a pattern has been explained by the increased fitness benefits of prior knowledge of the local environment for the sex that establishes territories (Greenwood, 1980; Gienapp & Merilä, 2011). This stronger territoriality and site fidelity in males observed in a reproductive context could lead to a similar tendency in the wintering areas, resulting in a higher predisposition for site-fidelity even in a gregarious (wintering) context. On the other hand, sexual dimorphism or behavioural differences between sexes may imply different ecological requirements during winter, such as differences in diet (e.g., Alves et al., 2013) or roost selection (e.g., Donázar & Feijóo, 2002). These differences could also drive sexual differences in site fidelity if required resources are heterogeneous in space. However, sexual differences in winter site fidelity have seldom been recorded in birds, being mostly restricted to species that frequently move during winter following spatial variation in food resources or that have family group composition affect their winter distribution (e.g., Roberts & Cook, 1999; Wunderle et al., 2014). Studies in highly site-faithful long distance migrants, such as waders, have not shown sexual differences thus far (e.g., Warnock & Takekawa, 1996; Catry et al., 2012).

Over the years, waders have been the focus of many individual-based studies, providing a significant source of information regarding site fidelity and individual movements. When faced with short-term changes in habitat quality/availability, some waders seem to be able to respond by moving up to over 100 km to more favourable areas (e.g., Kirby & Lack, 1993; Takekawa et al., 2002; Van Gils et al., 2006; Lourenço et al., 2010), while others continue to use degraded/changed habitats (Connolly & Colwell, 2005; Taft, Sanzenbacher & Haig, 2008) apparently due to limited ability to increase or change home ranges (Taft, Sanzenbacher & Haig, 2008). Understanding the level of site fidelity exhibited by individuals at a small spatial-scale (e.g., among sites within a single wetland) when compared with the continent-wide scale of their migration to and from their high-Arctic breeding sites (Loonstra, Piersma & Reneerkens, 2016) and, by opposition, the predisposition to disperse over increasing distances under stable conditions may provide some insights into the general ability of wader species to explore potential new areas or respond to short-term changes in their environment.

Sanderlings are common and widespread waders in coastal areas worldwide (Grond et al., 2015; Conklin et al., 2016). They are mainly associated with coarse sediment habitats, such as sandy beaches and sand or muddy-sand flats, which are particularly prone to quick changes through sediment migration and coastal erosion (Pethick, 2001) and often affected by human disturbance (e.g., Burger & Gochfeld, 1991; Thomas, Kvitek & Bretz, 2003). Sanderlings have been described as very site faithful and seem to have small home ranges at both wintering and staging areas (review in Reneerkens et al., 2009). In this study we use a dataset of observations of colour-marked sanderlings wintering in and around a large estuarine wetland to compare the level of winter site fidelity, within a single winter and among multiple winters, between sexes and age groups, and estimating the probability of these birds naturally dispersing over increasing distances under relatively stable conditions.

Figure 1 Map of the study area and proportion of birds sighted in each site that were either only sighted locally or also sighted in other sites.

(A) refers to the average per winter (4 winters for all sites except Caparica. For each we only had sightings of colour-marked birds in 2012–13). (B) refer to all winters combined. Each site is represented by a different colour (Alcochete: red, Samouco: yellow, Seixal: green, Caparica: blue, Oeiras: pink) and in the pie charts the proportion of birds only sighted locally have the colour of that site and the proportions of birds also sighted elsewhere have the colours of the other sites in which they were sighted. Number of marked individuals (n) that were detected at each site during the time span of the study: nAlcochete = 88, nSamouco = 42, nSeixal = 39 and nCaparica = 6.

Methods

Study area

Field work was carried out in the Tagus estuary, Portugal, one of the largest tidal wetlands in Western Europe, and comprised five sites known to harbour sanderlings during winter. Three of the sites are located within the estuarine area (Alcochete, Samouco and Seixal), and two are located on the oceanic coast near the mouth of the Tejo river (Caparica and Oeiras; Fig. 1). The minimum distance between a pair of sites is 3.1 km (Alcochete and Samouco), and the maximum is 29.8 km (Alcochete and Oeiras; Fig. 1). All study sites include foraging and roosting areas, the former located in the intertidal flats and the later situated above the high water mark. In Alcochete birds also roost on a saltpan located near the beach, which was also routinely monitored during the study. All sites harbour sanderling flocks throughout the winter with average counts of 75–126 individuals per site (more details regarding the study area in Lourenço et al., 2015).

Study population and data analysis

A total of 374 sanderlings were captured and ringed with individual colour-ring combinations in all three sites within the Tejo estuary in five consecutive non-breeding seasons between 2008/9 and 2012/13. Some birds were caught during late August and thus could still have been passage migrants, but since the re-sighting data refers only to the wintering period, this will not affect our winter site fidelity analysis. At capture, all birds were aged based on plumage characteristics, being classified as either first winter (hereafter referred as juveniles) or adults. Biometric data (wing length and tarsus length measured to the nearest mm; bill length and total head length measured to the nearest 0.1 mm) were also collected for most individuals to determine their sex. Sex of measured birds was determined based on a function derived from a generalised linear model (GLM) of biometric data from 990 molecularly sexed sanderlings captured in Mauritania and Ghana in a concurrent study (see Appendix S1 for details). Since juvenile sandpipers can have shorter wing lengths than adults (Yosef & Meissner, 2006), which is also true for sanderlings captured in the Tejo estuary (juveniles: 125.0 ± 0.3 mm, n = 89; adults: 127.1 ± 0.2 mm, n = 212; t299 = 5.03, p < 0.001), we only sexed adult birds through this method as wing is a key biometric sexing parameter (see Appendix S1). The GLM correctly assigned the sex of a large proportion of the 990 birds used to derive the model (84%; see Appendix S1), but to minimize the risk of incorrect assignments all birds with a sex assignment probability below 75% (i.e., 0.25 ≤ P(male) ≤ 0.75; n = 24) were excluded from the sex-related fidelity analysis. This resulted in 102 sexed birds, 64 males and 38 females.

From 2009/10 to 2012/13, during the sanderling wintering period (November–March), the study sites were visited frequently by us and many volunteer observers and the presence and identity of colour-marked birds was recorded. A total of 317 marked individuals were sighted, including 302 birds ringed locally and another 15 marked elsewhere (Greenland, Iceland and The Netherlands). The full dataset included 2,358 sightings. Since the dataset includes many sightings obtained during haphazard visits of volunteers (21% of all individual sightings used in the analysis), we do not know whether the searching effort was similar for all sites, because volunteer observers will not have contacted us when they did not find a colour-ringed sanderling during their visits. We also suspect that the sites outside the estuary are likely to have been visited less often than the sites located within the estuary. For the same reason, sightings are not evenly spread over each winter and over different winters (Table 1).

Table 1 Details on the data collected for the 160 marked sanderlings used in this study, including the number of individuals that were marked and sighted each winter (cumulative number in parenthesis), the number of sightings per individual each winter (average ± SE, range in parenthesis) and the average number of visits per site each winter.

Number of visits refers only to visits when at least one marked bird was sighted as there is no way of estimating the number of visits made by volunteers when no sightings were reported. Since marked birds were only detected in Caparica in 2012/13, the average only includes this site for that winter.

	2009/10	2010/11	2011/12	2012/13	
Marked individuals	44 (44)	28 (72)	42 (114)	46 (160)	
Sighted individuals	29	33	71	124	
Sightings per individual	4.0 ± 0.2 (3–7)	3.9 ± 0.2 (3–8)	4.4 ± 0.2 (3–14)	5.6 ± 0.23 (3–17)	
Visits per site	5.7 ± 1.2 (3 sites)	7.0 ± 1.5 (3 sites)	15.7 ± 1.6 (3 sites)	19.3 ± 4.0 (4 sites)	

In order to minimize the risk of having incorrect readings affecting our analyses, we limited our analysis to include birds that had been sighted a minimum of three times in a given area/year which resulted in 160 individuals (36 juveniles and 124 adults) for which we had a total of 1249 sightings (Table 1). For 101 of these individuals we had data over multiple years (65 seen in 2 winters, 28 seen in three winters and 8 seen in 4 winters). For each study site we calculated the proportion of individuals that were recorded only locally, and the proportion also recorded in other sites, both within a single winter (using the average of all four winters) and over multiple winters. To evaluate how the probability of dispersal varies with distance we used Generalized Linear Mixed-effects Models (GLMM) with logit link functions to relate a binomial variable indicating whether an individual sighted at a given site was later seen at each of the other sites, or only seen at the first site (movement distance = 0 km), to the distance among each pair of sites. Individual was used as a random factor in the GLMMs. This analysis was performed to assess inter-annual site fidelity (i.e., data from multiple winters), using all available sightings of each individual, and also intra-annual site fidelity in which case we used data available for each individual each year and used year as covariate. Age and sex were not included in this analysis because no juvenile birds were sexed (see above). The GLMM analysis were performed using package lme4 v1.1-11 in R (R Core Team, 2014).

In order to test differences in site fidelity between sexes and age classes, as well as differences in the site fidelity within a single winter and over multiple winters we calculated a site fidelity index (SFI) described by Catry et al. (2012). This index takes into account the number of sites used, the number of observed changes between sites and the total number of sightings for each individual: SFI=1−ni−1n−1×pioi−1

where ni is the number of sites used by individual i, n is the total number of sites surveyed, pi is the observed number of changes between sites performed by individual i and oi is the total number of sightings of individual i. SFI ranges from zero (no site fidelity) to one (complete site fidelity). For each individual, SFI values were calculated within each winter (intra-annual SFI) and for all winters combined (inter-annual SFI).

To investigate sex differences in site fidelity, we compared both intra-annual and inter-annual SFI values for males and females. For birds with data for multiple winters we used the average intra-annual SFI, and in both cases we used Mann–Whitney tests to compare SFI scores. Two approaches were used to compare site fidelity among age classes. One compared the intra-annual SFI values for juvenile and adult birds for the winter when each individual was ringed/aged, through a Mann–Whitney test; the other used only birds ringed as juveniles and sighted over multiple winters, in which case we made a pair-wise comparison of intra-annual SFI values calculated for each juvenile in their first winter and in the subsequent winter (when adult) with the most sightings of that particular individual, using a Wilcoxon matched pairs test.

Finally, in order to compare site fidelity within the same winter and over multiple winters, we made a pair-wise comparison of intra and inter-annual SFI values for all adult birds sighted over multiple years using a Wilcoxon matched pairs test. There could be some biases when comparing inter-annual SFI of individuals with different numbers of sightings in each winter, but this problem would only affect comparisons among individuals and not pair-wise comparisons for each individual. Note that birds ringed as juveniles were excluded from this analysis because the tests described above showed that sanderlings are less site faithful in their first winter (see below). Data are presented as average±SE.

Results

The majority of individuals (93.6 ± 0.7%, n = 4 winters with an average 73 ± 25 individuals/winter) were always sighted at the same site within a given winter (Fig. 1). The same pattern was observed for multiple winters (Fig. 1) but the proportion of birds that were only observed at a single site decreased with the number of winters considered, with 81.5% (n = 65) for 2 winters, 67.9% (n = 28) for 3 winters and 62.5% (n = 8) for 4 winters. Bird movements were more likely to occur between sites in close proximity, namely between Alcochete and Samouco, between Samouco and Seixal and between Seixal and Caparica (Fig. 1).

In fact, the GLMMs indicated that the probability of movement was negatively affected by distance, with lower probabilities of dispersal between sites further away both within one winter (β = − 0.609 ± 0.071, z = 8.61, p < 0.001) and over multiple winters (β = − 0.278 ± 0.026, z = 10.57, p < 0.011; Fig. 2). In the analysis for a single winter the covariable ‘year’ had no effect on the probability of movement (z = 0.45, p > 0.5). In fact, there was only a single case of dispersal between sites located at a distance of over 20 km. No marked birds were ever detected at Oeiras even though the site was frequently used by non-marked sanderlings.

Figure 2 Effect of distance between sites on the probability of movements occurring among them.

The grey symbols and GLMM regression line refer to a single winter (average of all four winters), while the black symbols and line refer to multiple winters. Each dot shows the average probability of movement for a given distance. For each pair of sites, the number of marked individuals ranged from 40 (Seixal-Caparica) to 117 (Alcochete-Samouco).

Females exhibited significantly lower intra-annual SFI values than males (intra-annual SFImale = 0.999 ± 0.001, n = 64; intra-annual SFIfemale = 0.991 ± 0.005, n = 38; Z = 2.15, p < 0.05; Fig. 3A), while there was no significant difference in inter-annual SFI values between sexes (inter-annual SFImale = 0.989 ± 0.004, n = 42; inter-annual SFIfemale = 0.979 ± 0.007, n = 31; Z = 1.40, p > 0.1; Fig. 3B). Juvenile birds had significantly lower intra-annual SFI values than adults in the winter when they were ringed/aged (intra-annual SFIadult = 0.997 ± 0.003, n = 126; intra-annual SFIjuvenile = 0.975 ± 0.009, n = 34; Z = 3.56, p < 0.001; Fig. 3C). The intra-annual SFI values of birds ringed as juveniles increased in subsequent winters (intra-annual SFIfirst winter = 0.977 ± 0.009, n = 24; intra-annual SFIsubsequent winter = 0.995 ± 0.005, n = 24; Z = 1.99, p < 0.05; Fig. 3D). For adult birds, inter-annual SFI values were significantly lower than intra-annual SFI values (intra-annual SFI = 0.995 ± 0.003, n = 101; inter-annual SFI = 0.985 ± 0.004, n = 101; Z = 2.59, p < 0.01; Fig. 3E).

Figure 3 Comparison of Site Fidelity Index (SFI) values between sexes (A and B) and age classes (C and D), and between a single winter and multiple winters (E).

(A) represents average intra-annual SFI values for males (n = 64) and females (n = 38); (B) represents average inter-annual SFI values for males (n = 42) and females (n = 31); (C) presents average intra-annual SFI values for adults (n = 126) and juveniles (n = 34) in the winter when they were ringed/aged; (D) presents average intra-annual SFI values of 24 birds ringed as juveniles when considering their first winter (juveniles) and a subsequent winter (adults); and (E) presents average intra and inter-annual SFI values of 101 birds ringed as adults. The black dots represent the mean, the boxes represent standard errors and the whiskers represent the range. * p < 0.05, ** p < 0.01, *** p < 0.001.

Discussion

Despite some movements among sites, our data show high levels of short-scale winter site fidelity in sanderlings wintering in the Tejo estuary region, especially if we consider that the study sites are located in close proximity, with a minimum distance of just 3.1 km between sites. Such high level of site fidelity is remarkable for a long-distance migrant accustomed to fly thousands of kilometres (several birds sighted in the Tejo were marked in breeding sites in northern Greenland, roughly 4000 km away, J Reneerkens (2016, unpublished data), and which disperses over a huge latitudinal range during the winter (in the East Atlantic Flyway at least 58°N to 23°S, e.g., Loonstra, Piersma & Reneerkens, 2016), but it is in line with previous studies both on sanderlings (Evans, Breary & Goodyer, 1980; Myers et al., 1990; Gudmundsson & Lindström, 1992) and other waders (e.g., Burton & Evans, 1997; Leyrer et al., 2006; Catry et al., 2012).

However, sanderlings do visit other sites and, after four winters, roughly one third of all birds had already been sighted in at least two sites. The very high intra-annual SFI values and the significantly lower inter-annual SFI values suggest that movements rarely occur during winter, but rather that birds may change wintering location between winters. These data also support the idea that movements are probably not influenced by stress caused by our catching events. Although we did not have the statistical power to analyse how the probability of movement changes over the course of the winter, the majority of sanderlings seem to select a wintering site early after arrival from their High Arctic breeding areas and remain faithful to that site over the rest of the winter. The occupation of wintering sites in Ghana seems to be regulated by a buffer effect, indicating that when sanderlings’ wintering sites have reached their carrying capacity, individuals were forced to use other non-preferred sites (Ntiamoa-Baidu et al., 2014). Hence, the first arriving birds in autumn are more likely to return to previously used winter sites. Given that juveniles arrive later in the wintering areas than adults (e.g., Lemke, Bowler & Reneerkens, 2012), the juveniles would be more prone to be forced to non-preferred sites.

Both age and sex influenced the level of site fidelity of sanderlings. However, since we did not sex juveniles, we cannot determine if there is any interaction between these two variables. The lower site fidelity exhibited by juveniles seems to be in line with evidence from other avian species where younger birds show larger home ranges (Warnock & Takekawa, 1996) and are more likely to switch roosting and feeding locations (Rehfisch et al., 1996) during winter. In fact, an experimental study involving displacement of dunlins Calidris alpina showed that site fidelity of juveniles seems to increase even within their first winter. Birds that were moved later in the winter were more likely to show “homing” behaviour (Baccetti et al., 1999). Although this age effect is not observed in all bird species (e.g., Diefenbach, Nichols & Hines, 1988; Fox et al., 1994; Monsarrat et al., 2013), such differences may reflect the fact that, as age increases, birds may be using accumulated knowledge of site specific characteristics or past experiences to develop preferences for particular sites, while dominance can also play a role with juveniles being more easily displaced by older, more dominant individuals (e.g., Groves, 1978). Consequently, older animals with presumably greater knowledge of wintering site characteristics and more past experiences may show increased fidelity to wintering sites which they found to be profitable and/or safe, similarly to what is known for breeding site fidelity (e.g., Serrano et al., 2001).

Unlike previous studies, which did not find sexual differences in winter site fidelity of waders (e.g., Warnock & Takekawa, 1996; Catry et al., 2012), we detected significantly lower site fidelity in females within a single winter. It must be noted that this difference is quite small, with the vast majority of birds of both sexes remaining faithful to their wintering site throughout the winter. This small difference may have hampered its detection in previous studies using different methods. In the breeding areas, many bird species show higher site fidelity in the sex that establishes territories (Greenwood, 1980; Gienapp & Merilä, 2011). Sanderlings can also be territorial in winter (Myers, Connors & Pitelka, 1979) so we cannot rule out that territoriality may play a role here, but since sanderlings are also gregarious and frequently change group composition within the same foraging site (Myers, 1983; Roberts & Evans, 1993) this seems unlikely. Lower site fidelity could indicate that females have less benefits from prior knowledge of their foraging or roosting sites, or more benefits from exploring alternative foraging sites if for instances their preferred prey has a different distribution than that from the males (Alves et al., 2013). However, since sanderlings shows little sexual dimorphism (Engelmoer & Roselaar, 1998; supplementary information to this study), there is no evidence for differences in energetic costs or foraging behaviour that could drive the observed sexual difference. In fact, the few cases in which sexual differences in winter site fidelity have been reported in bird species which make frequent movements during winter following changes in food availability, or species in which pair bonds or family group composition affect wintering distributions (e.g., Roberts & Cook, 1999; Wunderle et al., 2014). Further studies will be required to confirm whether the sexual trend in winter site fidelity we observed is a general feature in sanderlings, and potentially other site-faithful long-distance migrants.

Overall, and despite the discussed sexual and age differences, sanderlings are highly site faithful to their wintering sites. Also, the observed movements are mostly on a small scale and there was only a single case of a bird moving between sites located further than 20 km apart. In fact, Oeiras was never used by any marked bird and this site is located 20.5 km away from the nearest estuarine site where birds were marked (Seixal), while Caparica which is just 14.5 km away from the same site was used by 6 marked individuals. This suggests that 15–20 km is indeed the maximum distance sanderlings are likely to move among wintering sites under stable conditions. As is the case for most studies focusing on animal dispersal (Nathan et al., 2003), our study is also hampered by the impossibility to sample all potential areas past a limited distance. The maximum distance between our study sites was 29.8 km, which is a very considerable distance and larger than in many similar studies (e.g., Evans, Breary & Goodyer, 1980; Groen, 1993; Burton & Evans, 1997; Leyrer et al., 2006), but we still may be failing to detect movements to areas located further away. There is also the possibility that within this 30 km radius birds could use other sites within and in the vicinity of the estuary that were not monitored, although the relatively rare occurrence of sandy substrates in the Tejo estuary and the known preference sanderlings show for these substrates suggest this is unlikely (Lourenço et al., 2015). Also, since ringing only took place in the three sites within the estuary, and not on the furthest sites located outside the estuary, while the latter were probably visited less often, there might be a bias against detecting movements over larger distances.

Still, even when considering these biases and limitations, there is a clear pattern of site fidelity and decreasing chance of movements over increasing distances. The few birds ever detected outside the estuary originated from the sites nearer the mouth of the estuary (mainly the nearest, Seixal), and never from the site located the furthest (Alcochete) even though this was the site with the largest number of individual birds ringed and sighted. The few sightings elsewhere along the Portuguese coast of sanderlings that were colour-marked in other countries than Portugal, suggest similar site fidelity: from 37 sightings of 12 individuals (average sightings per individual: 3.1 ± 2.0), each observed in a single winter, only one was observed at two locations which were separated by less than 4 km (Póvoa de Varzim 41°23′N/8°46′W and Vila do Conde 41°21′N/8°45W).

Altogether, our data suggest that sanderlings in and around the Tejo estuary rarely move among wintering sites located at distances of more than 15 km. Such a level of site fidelity during winter is likely to confer advantages related to previous knowledge of foraging locations, potential predation risks, and social interactions with other individuals using the same areas (e.g., Alerstam, 1990; Snell-Rood & Cristol, 2005), but may also limit the ability of these birds to detect new areas of favourable habitat that may become available for colonization (Matthiopoulos, Harwood & Thomas, 2005) or even of avoiding the consequences of any degradation to their traditional wintering sites (e.g., Porzig et al., 2014).

In the Netherlands, individually recognisable sanderlings have been observed to move over tens of kilometres in response to a sudden increase in food availability generated by a storm washing up large amounts of American Jack knife clam Ensis americanus on the shore (J Reneerkens, 2016, unpublished data), suggesting that sanderlings are able to react to the availability of new foraging opportunities. Also, in the Solent estuary, UK, marked sanderlings seem to move regularly among feeding sites located 15–20 km apart (PM Potts, 2015, unpublished data). There is no evidence that any of the three areas we studied within the Tejo estuary suffered significant changes during the course of this study, so we cannot predict whether the birds would move if there was any decline in habitat quality, but these anecdotal observation from The Netherlands and UK suggest that they are able to move to a larger extent than we observed in the Tejo. However, at Caparica, the beaches are known to be suffering from severe coastal erosion and are estimated to be retreating inland at an average rate of 7 m year−1 as a result of river damming and dredging, inadequate coastal management and urban pressure, and sea-level rise (Ferreira & Matias, 2013). These beaches only persist there due to intense artificial sediment nourishment (Ferreira & Matias, 2013) which can result in severe ecological impacts and loss of biodiversity (Schlacher et al., 2007). In fact, data collected in parallel to this study indicate both food availability and sanderling intake rates are currently lower in Caparica than in the sites within the estuary (Lourenço et al., 2015). Despite this, sanderlings continue to use this site and the few birds ringed in the estuary that were sighted in Caparica continued to use the site throughout the winter. This may indicate that site fidelity is a stronger driver of habitat selection for sanderlings than foraging habitat quality (as far as we can measure it) with potentially negative consequences for birds using areas suffering fast human-mediated changes such as the case of many coastal areas worldwide (Zhang, Douglas & Leatherman, 2004; Schlacher et al., 2007).

Supplemental Information

Data S1 Site fidelity data for individually-marked sanderlings wintering in the Tejo

Click here for additional data file.

Appendix S1 Appendix: Biometric Sexing of sanderlings (Calidris alba)

Click here for additional data file.

We would like to thank all the observers who voluntarily sent us sightings of our colour-marked birds, including several members of the Farlington Ringing Group who routinely visit the Tejo estuary and provided great help in catching, marking and re-sighting sanderlings. Particularly, Ruth Croger, Anne de Pottier and Mark Fletcher visited our study areas each winter and were an important help in the field. We would also like to thank Afonso Rocha, Sara Pardal and Miguel Braga for help in catching and marking sanderlings.

Additional Information and Declarations

Competing Interests

Author Contributions

Animal Ethics

Data Availability

Peter M. Potts is an employee of the Farlington Ringing Group. The other authors declare that there are no competing interests.

Pedro M. Lourenço and José A. Alves conceived and designed the experiments, performed the experiments, analyzed the data, contributed reagents/materials/analysis tools, wrote the paper, prepared figures and/or tables, reviewed drafts of the paper.

Jeroen Reneerkens and Teresa Catry conceived and designed the experiments, performed the experiments, contributed reagents/materials/analysis tools, wrote the paper, prepared figures and/or tables, reviewed drafts of the paper.

AH Jelle Loonstra wrote the paper, reviewed drafts of the paper.

Peter M. Potts performed the experiments, contributed reagents/materials/analysis tools, reviewed drafts of the paper.

José P. Granadeiro conceived and designed the experiments, analyzed the data, contributed reagents/materials/analysis tools, wrote the paper, prepared figures and/or tables, reviewed drafts of the paper.

The following information was supplied relating to ethical approvals (i.e., approving body and any reference numbers):

The protocol was approved by the responsible ethical and legal authority, the Portuguese Institute of Nature Conservation and Forests (ICNF) and performed under official permits 385/2013/CAPT, 386/2013/CAPT and 387/2013/CAPT.

The following information was supplied regarding data availability:

The raw data has been supplied as a Supplemental File.

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
