# Peer review of "Influence of age and sex on winter site fidelity of sanderlings Calidris alba"

_PeerJ, doi:10.7717/peerj.2517_

## Round 0.1 · original submission · Major Revisions

· Academic Editor

Major Revisions

As usual for manuscripts submitted for publication as Original Paper, your paper has been evaluated by two peer reviewers, one of who (#2) recommend minor corrections, but (#1) recommend rejection. Their basic conclusions differ substantially. Reviewer #1 is clearly more critical and also expressed some major concerns with your work, basically recommending to reject the publication. After carefully pondering the reviews, I arrive at this decision: The manuscript does generally merit publication but it is still not acceptable in its revised form and needs a further, rather minor revision based on the reviews. You need to take care of all concerns expressed by both reviewers, and when you submit the revised manuscript, please also provide a detailed RESPONSE. Please state point-by-point which changes you have made in response to the reviews especially those concerns of rev #1 and where and why you have refused to follow a particular suggestion or you consider it. If the accordance between the changes and the reviewers' requests is sufficiently transparent, no further reviewing will be needed. Please send an annotated version “Marked Manuscript” in which the changes you made are clearly indicated highlighting the modified text.
If you are prepared to undertake the work required, I would be pleased to reconsider my decision and eventually accept a revised manuscript for publication

Reviewer 1 ·

Basic reporting

Submission adheres to PeerJ policies.
It is written in clear and unambiguous English.
Introduction is sufficient and relevant literature listed.
Structure follows templates.
The hypothesis is relevant.
Data is provided.
While revising tables figures and appendix I found several minor inconsistencies that could be improve.
Table 1:
I suggest aligning text from the first column to the upper part of the box. This will make reading the table easier. I noticed that even if volunteers visited all 5 sites every year you do not have information of the 5 sites every year this is a strong limitation of your work as it is.
Figures:
Figure1: Small square map of the Iberian Peninsula should be noted as panel A. The scale of the small map is not included. You should explicitly explain that the small grey rectangle correspond with the study area. Panel A should be panel B and panel B should be panel C. Figure would not be readable when printed in black and white and is not readable for colour blind people. Figure caption is misleading. If panel A refers to average proportion of birds sighted for each one of the 4 years at each one of the 5 locations, shouldn´t you plot one pie chart by year in each location? At the end of the caption you refer to the number of marked individuals detected at each site, is this a total? Average over years?
Figure 3: Top line should be panel A and B, middle should be panel C and D, and panel E is not referred in the second line of the text, please include.
Appendix:
Table S1: Caption states ±sd appears between brackets in the table, it does not. The table is not self-explicative. Please explain in the caption what do N, range and t mean. I suggest to be consistent in the number of decimals you use through the manuscript ( notice that within this table you use 0,1 and 3 decimals). Toe from Tarsus Toe should be in lowercase.
Table S2: Table is not self-explicative please explain what AIC, P, DF mean. You state: lower values indicate a better fit. Values of what? Model selection procedures are not explained in the manuscript please do. See my previous comments on model selection.
Notice that you stated abbreviations for some of your covariates (Tarsus, T; Bill length, B and so on). Maybe the model selection table would be more easily readable if you use the same notation.

Experimental design

They clearly identify the research question and why it is meaningful.
In my opinion the used methodology does not allow to draw they conclusions they reached.
Their overall result is that despite they find differences between sexes those differences are small and easily negligible. Their results are in accordance with previous studies as they suggest in the main text. Yet probability of dispersal is calculated by GLMM analyses which do not account for recapture probability instead of using a capture-recapture modelling framework. In my opinion this article would be much more robust if they estimated probability of movement from a capture-recapture modelling perspective. In the main text they state that the roosting sites located outside the estuary were less likely to have been monitored therefore probability of recapture there is lower. In fact, data from Caparica is only available for 2 of the 4 years. No recaptures from Oeiras are available, this could be a consequence of lower use of this roosting site or a result of sampling asymmetry. Since they do not provide clear information about the amount of recaptures that you have for each of the sites by year therefore is hard to know if they will be able to perform such analyses. I believe the data and the analyses they provided in terms of dispersal probability are not sound enough to draw their final conclusion. Authors use SFI with which methodology I´m not familiarized therefore I’m not sure if my comments about it would be very useful.
Besides the fact that I would like see this data analysed from a capture-recapture perspective, the methods section lacks explanation of the model selection procedures. You might find useful Burnham and Anderson (2002) book.
The discussion section could use some extra work. I noticed that until line 328 the authors suggest that limited movement between wintering areas might result in decreased ability to adapt to changing conditions. Yet from line 329 they present a series of unpublished data that suggest 1) That sanderlings are actually able to move longer distances than the ones reported in the article, and 2) to respond to very concrete spatial-temporal changes. I found this late part of the discussion rather confusing.
Moreover, I might have missed it but I think you do not discuss why there is no resight in Oeiras ( notice that it is more or less to the same distance to Samouco than Caparica to Alcochete).
Moreover:
Tittle states: Influence of age and sex on winter site fidelity of sanderlings Calidris alba. However, in your results and especially in your discussion session you emphasize that there is no influence of age and sex on winter site fidelity. I suggest you to rephrase your title accordingly.
Abstract: The abstract includes methodological details (such as the sexing procedure) that could be removed ( in fact authors moved this part from the main text to the appendix).

Validity of the findings

As I previously stated I believe the data and the analyses they provided in terms of dispersal probability are not sound enough to draw their final conclusion. I have no expertise in terms of SFI so I refrained myself from commenting that part.
I encourage the authors to keep working on their manuscript until they achieve publication.

·

Basic reporting

No Comments

Experimental design

No Comments

Validity of the findings

No Comments

Additional comments

This is a nice study of site fidelity in sanderlings Calidris alba. This manuscript present an analysis based on observations of marked birds in different sites. It is a clear example of the value of citizen science for scientific research. Based on their results, authors suggest that sanderlings are site-faithful and that this could be a problem if habitat quality changes. Regarding methods, I am not aware if the SFI is widely used in this kind of studies but I have suggested some other alternatives (which I consider more parsimonious) in the manuscript. Also, it is not clear what proportion of the total distance that this species migrates is covered in this study. I would appreciate if authors could add more information about that.
The text is written in clear English and the results are well presented in Tables and Figures. In my opinion all the weak points of the study are very well discussed. Nevertheless, you may want to see my specific comments on the manuscript and take them as opportunities of improvment.

---

## Round 0.2 · accepted · Accept

· Academic Editor

Accept

After evaluating your resubmission and your responses to my comments on the previous version, I am pleased to tell you that I arrived at the decision that all issues raised in the peer review process are now settled and your revised ms is ready for being published.